# An elementary concentration bound for Gibbs measures arising in statistical learning theory

**Kelly Ramsay**                                                    *kramsay2@yorku.ca*
*Department of Mathematics and Statistics*
*York University*
*North York, ON M3J1P3, Canada*

**Aukosh Jagannath**                                               *a.jagannath@uwaterloo.ca*
*Department of Statistics and Actuarial Science*
*University of Waterloo*
*Waterloo, ON N2L3G1, Canada*

**Shoja'eddin Chenouri**                                           *schenouri@uwaterloo.ca*
*Department of Statistics and Actuarial Science*
*University of Waterloo*
*Waterloo, ON N2L3G1, Canada*

**Reviewed on OpenReview:** *https://openreview.net/forum?id=ZInwrlkQ3f*

## Abstract

We present an elementary concentration bound for Gibbs measures whose log-likelihood is a function of the empirical risk. This bound controls the distance between samples from the (random) Gibbs measure and the minimizers of the population risk function. This bound is a generalization of a recent inequality developed by Ramsay et al. (2024). As a corollary, we obtain sample complexity bounds and bounds on the inverse temperature so that the samples are within a prescribed error of the population value. The latter bound on the inverse temperature is essentially sharp. We demonstrate our work on three canonical classes of examples: classification of two component mixture models, robust regression, and spiked matrix and tensor models.

## 1 Introduction

A basic task in learning theory is empirical risk minimization. Empirical risk minimization can be challenging: for example, the empirical risk can be non-convex or involve combinatorial constraints. There is a vast literature tackling these challenges from many perspectives.

One popular approach to circumvent the aforementioned issues is to (approximately) sample from a Gibbs measure whose log-likelihood is proportional to the empirical risk, where the proportionality constant, called the *inverse temperature* is a hyperparameter to be tuned. This perspective is the main motivation for the popular method of "simulated annealing" (Kirkpatrick et al., 1983). That said, in recent years, this perspective has motivated analyses for other algorithms such as Stochastic Gradient Langevin Dynamics (Welling & Teh, 2011; Raginsky et al., 2017; Zhang et al., 2017) and Stochastic Gradient Descent (Mandt et al., 2017; Cheng et al., 2020; Yu et al., 2021). Methods involving samples from Gibbs measures of this type have been used to tackle a broad range of problems in learning theory ranging from supervised and unsupervised learning (Aubin et al., 2018; Coja-Oghlan et al., 2018; Barbier et al., 2019), to inference problems with combinatorial structure (Jerrum, 1992; Gamarnik et al., 2021; Ben Arous et al., 2020), to differential privacy (McSherry & Talwar, 2007).

When studying such approaches there are two important, but distinct issues to consider: one is algorithmic in nature, namely determining the run time or rate of convergence of the sampling scheme, and one is statistical

in nature, namely determining the quality of a sample from the Gibbs measure as an estimator. We focus on the statistical aspects of the problem. For the algorithmic setting, see, e.g., (Mandt et al., 2017; Zhang et al., 2017; Raginsky et al., 2017; Green et al., 2015) and the references therein.

There are many deep analyses for specific problems, e.g., (Lelarge & Miolane, 2017; Perry et al., 2020; Jagannath et al., 2020; Bhattacharya & Martin, 2022). Statistical analyses taking a general perspective have focused mainly on asymptotic theory (Ghosal et al., 2000; Shen & Wasserman, 2001; Chernozhukov & Hong, 2003; Grünwald & Mehta, 2016; Syring & Martin, 2020; Bhattacharya & Martin, 2022; Bochkina, 2022). Another line of work concerns the population risk of the model sampled from the Gibbs measure (Aminian et al., 2021; Bu, 2024; Perlaza et al., 2024; Zou et al., 2024). In particular, PAC-Bayesian bounds, which are high probability bounds on the generalization error of the model sampled from the Gibbs measure, are an active area of research, see (Alquier et al., 2024) and the references therein. Furthermore, Bu (2024) considered the choice of inverse temperature. They derive the inverse temperature that minimizes the population risk of the model sampled from the Gibbs measure. We also consider the finite sample performance of these estimators, along with the choice of inverse temperature, however, our goal is conceptually different. We seek concentration bounds for these estimators with the aim of answering the following, naive question:

> How large should the inverse temperature, $\beta$, and the sample size, $n$, be to guarantee that a sample from the Gibbs measure is within a *distance t* of a minimizer of the population risk?

Observe that there is an importance conceptual difference between bounding the distance to the minimizer of the population risk—the problem considered here—and bounding the value of the population risk achieved by the estimator, the problem canonically studied in the PAC-Bayes literature. (Under certain additional assumption one may relate the two, e.g., if the population risk is known to be monotone in this distance.)

As mentioned above, in this paper, we provide an elementary, quantitative concentration bound on the distance between the sampled parameter and the true minimizer of the population risk. This bound depends on the accuracy of the prior, the inverse temperature, and the number of samples, thereby allowing us to provide a quantitative answer to the above question. As one might expect, this bound depends on a quantification of the trade-off between the two inherent sources of randomness in the problem: the "energy-entropy" trade off of the Gibbs measure and the randomness in the underlying data. As a direct consequence, we obtain a quantitative answer to the above question, namely bounds on the sample complexity and the inverse temperature required to obtain a prescribed error level with high probability.

Our bound is an extension of a concentration bound recently introduced by Ramsay et al. (2024) in the context of differential privacy. Indeed, their bound can be recovered by our bound. The goal of this paper is to illustrate how one can apply the main technical idea of Ramsay et al. (2024) to a broader class of problems in statistical learning beyond differential privacy. In particular, our main result has weaker conditions than that paper: Conditions 1 and 2 are the same in both papers. Condition 3 of Ramsay et al. (2024) implies Condition 3 in our paper by Talagrand's inequality for empirical processes. By relaxing this condition, the range of problems to which this lemma applies is dramatically increased. In plain language, Condition 3 of Ramsay et al. (2024) implies that changing one observation in the dataset to an arbitrary value cannot affect the empirical risk function greatly at any point in its domain. This is a common requirement in differential privacy, but not in the non-private setting. Here, we relax Condition 3 of Ramsay et al. (2024) so that the inequality can be used in a broad range of settings in statistical learning theory, beyond differential privacy.

We demonstrate our bound on three classes of learning problems: classification of a two component mixture model, robust regression, and inference for spiked matrix and spiked tensor models. (Finally, we note here that, as shown in Remark 6.2 below, the bound we obtain is essentially sharp.)

## 2 A concentration inequality

Let us begin by stating our main concentration inequality, which is a generalization of a concentration bound first shown by Ramsay et al. (2024). In particular, we present a bound which applies to learning problems beyond differential privacy through relaxing the main conditions. Suppose that we are given $n$ i.i.d. observations, $X_1, \ldots, X_n$, from a probability measure $\mu$ on a complete separable metric space $(\mathcal{X}, \tilde{d})$.

Given a loss function $\ell \colon \Theta \times \mathcal{X} \to \mathbb{R}$, we seek to estimate a minimizer of the *population risk*, namely $R(\theta) = \mathbb{E}_\mu \ell(\theta, X)$. Here $\Theta$ is the parameter space - the set of candidate minimizers. We assume that $(\Theta, d)$ is also a metric space. Our estimate is given by one draw from the Gibbs distribution,

$$\nu_\beta(d\pi) \propto \exp(-\beta \widehat{R}_n(\theta)) d\pi, \tag{2.1}$$

that is, $\tilde{\theta}_n \sim \nu_\beta$, $\pi$ is a prior on the unknown parameter, $\widehat{R}_n(\theta) = n^{-1} \sum_{i=1}^n \ell(\theta, X_i)$ is the empirical risk and $\beta > 0$ is a fixed hyperparameter called the *inverse temperature*. Note that if $\tilde{\theta}_n$ concentrates around a minimizer of the population risk, this generally implies bounds on the accuracy of other familiar estimates based on the above Gibbs measure. For instance, if $\tilde{\theta}_n$ concentrates around the minimizer of the population risk then we immediately get a bound on $\int \theta d\nu_\beta(\theta)$. Let $A^* = \{\theta : R(\theta) = \min_\eta R(\eta)\}$ denote the set of (global) minimizers of $R(\theta)$, which is assumed to contain at least one point (but may contain more). Our bound requires the following three conditions on the triple $(\ell, \mu, \pi)$.

**Condition 1.** The function $R(\theta)$ has a minimizer and $\int \exp(-\beta \widehat{R}_n(\theta)) d\pi < \infty$.

**Condition 2.** The function $R(\theta)$ is $L$-Lipschitz for some $L > 0$.

**Condition 3.** There exists $t_0 := t_0(n, \Theta), c_1 := c_1(n, \Theta), c_2 := c_2(\Theta) > 0$, such that for all $t \geq t_0$, it holds that

$$\Pr\left(\sup_{\theta \in \Theta} \left|\widehat{R}_n(\theta) - R(\theta)\right| \geq t\right) \leq c_1 e^{-c_2 n t^2}.$$

Condition 1 ensures the problem is well-defined. Condition 2 is a smoothness condition on the risk function. This can be relaxed to a uniform continuity condition if needed, see (Ramsay et al., 2024) for a similar modification. Condition 3 says that the empirical risk concentrates uniformly around the population risk at a sub-Gaussian rate. Condition 3 is in principle the hardest to check. It can be checked by various methods from concentration of measure, such as Talagrand's inequality for empirical processes (Talagrand, 1994) (see Section 4) or a Logarithmic–Sobolev inequality (see Section 3).

We also need the following important quantities to state our concentration result. For a set $E \subseteq \Theta$, let $B_r(E) = \{\theta : d(\theta, E) \leq r\}$. Specifically, the *minimum excess risk* is given by

$$\alpha(t) = \inf_{\theta \in B_t^c(A^*)} R(\theta) - \inf_{\theta \in \Theta} R(\theta). \tag{2.2}$$

It measures the minimum excess risk of a point $\theta$ that is at least $t$ distance away from the minimizing set. The calibration function is given by

$$\psi_\pi(\lambda) = \min_{t>0}[\lambda \cdot L \cdot t - \log \pi(B_t(A^*))]. \tag{2.3}$$

The calibration function is small when the risk is smooth and there is a small neighborhood of the minimizing set on which the prior is non-negligible. Lastly, the rate function of the prior

$$I(t) = -\log \pi(B_t^c(A^*)) \tag{2.4}$$

measures the rate of decay of the tails of the prior. Our main result is then as follows.

**Theorem 2.1.** *Suppose that the triple $(\ell, \mu, \pi)$ satisfy Conditions 1–3 for some $t_0, L, c_1, c_2 > 0$. Then, for any $\beta^{-1} I(t) \vee \alpha(t) \geq 2t_0$, we have*

$$\Pr\left(\max_{\theta \in A^*} d(\tilde{\theta}_n, \theta) \geq t\right) \leq c_1 e^{-c_2 n\left[\frac{1}{\beta} I(t) \vee \alpha(t)\right]^2/4} + e^{-\beta \alpha(t)/2 - I(t)/2 + \psi_\pi(\beta)}. \tag{2.5}$$

The upper bound given in Theorem 2.1 consists of two terms. The first term represents the error from using the empirical risk $\widehat{R}_n(\theta)$ to approximate the population risk function $R(\theta)$, and the second term represents the error from using a draw from the Gibbs measure, rather than the minimizer of the empirical risk. For the sake of discussion, let us call these *the sampling error term* and *the energy-entropy term*, respectively.

Theorem 2.1 can be used to prove bounds on $\beta$ under which the estimate $\tilde{\theta}_n$ achieves optimal sample complexity. To see this, observe that by Markov's inequality, Theorem 2.1 also implies a high-probability bound on the probability of $\nu_\beta(B_t(A^*)^c)$. Now, for quantities $a, b$, we write $a \lesssim b$ when there is a universal constant $C > 0$ such that $a \leq Cb$. Next, when $n \gtrsim \log(1/\gamma)/c_2\alpha(t)^2$ the sampling error term is bounded by $\gamma$. This bound on $n$ yields an upper bound on the sample complexity which is often optimal, up to logarithmic factors, see the example applications in the sections that follow. Then, if one chooses $\beta$ such that $\beta \gtrsim (\psi_\pi(\beta) \vee \log(1/\gamma))/\alpha(t)$ the energy-entropy error term is also bounded by $\gamma$. Then, the problem amounts to bounding $-\log(\pi(B_t(A^*)))$ for small $t$, see Lemma 6.1. It is interesting to note that bounding $-\log(\pi(B_t(A^*)))$ is akin to the conditions on the prior necessary to apply the results given by Shen & Wasserman (2001); Syring & Martin (2020).

Theorem 2.1 requires that $\beta^{-1}I(t) \vee \alpha(t) \geq 2t_0$, the impact of which depends on the application. For instance, in all our example applications, it ends up being unimportant. When applying Theorem 2.1 we use the fact that $\alpha(t) > t_0$ implies $\beta^{-1}I(t) \vee \alpha(t) > t_0$, and then the bound $\alpha(t) > t_0$ is either always satisfied, or is implied by other bounds on $n$ required to achieve optimal sample complexity.

*Remark* 2.2. Condition 3 can be weakened at the expense of slower concentration rates. For example, if instead the supremum of the centered empirical risk process has a large deviations upper bound with rate $n$, i.e.,

$$\Pr\left(\sup_{\theta \in \Theta} \left|\widehat{R}_n(\theta) - R(\theta)\right| \geq t\right) \leq c_1 e^{-c_2 n J(t)},$$

for some monotone rate $J$, then the sampling error term in equation 2.5 becomes $c_1 \exp(-c_2 n J(\beta^{-1}I(t) \vee \alpha(t))/4)$.

*Remark* 2.3. Note that, when $\beta$ becomes large, we expect the error of $\tilde{\theta}_n$ to be similar to that of the true minimizer of the empirical risk, say $\hat{\theta}_n$. This is reflected in equation 2.5, where, under a weak condition, the energy-entropy term disappears as $\beta \to \infty$. The condition is that $\beta \to \infty$ faster than $\psi(\beta)$, which holds when the prior has sufficient density on the minimizing set and the risk is smooth enough. Going further, taking $n \to \infty$, we have that equation 2.5 implies that $\tilde{\theta}_n$ is weakly consistent for $\theta$.

## 3 Classification for a two component mixture model

As our first example, consider the basic task of supervised classification of a two component Gaussian mixture model via a single-layer neural network. Suppose that we are given data of the form $\{(X_i, Y_i)\}_{i=1}^n$ where $Y_i$ are i.i.d. Rademacher random variables and $X_i = Y_i Z_i$ where $Z_i \sim \mathcal{N}(v, I)$ are i.i.d. and $v$ is a fixed, but unknown, vector satisfying $\|v\| = 1$. We view $X_i \in \mathbb{R}^d$ as the features and $Y_i \in \{-1, +1\}$ as the class labels and our goal is to develop a classifier for this problem. The standard approach is to develop such a classifier via a single-layer neural network. More precisely, we take the loss function to be

$$\ell(\theta, (X, Y)) = -\log \sigma(Y\langle X, \theta\rangle),$$

where $\sigma(x) = (1 + \exp(-x))^{-1}$ is the usual sigmoid function and $\theta \in \mathbb{S}^{d-1}$. The corresponding classifier is then obtained by a simple thresholding: $\widehat{y}(x) = (2 \cdot \mathbb{1}(\sigma(\langle x, \theta\rangle) > 1/2) - 1)$.[1] For this example, we take $\pi$ to be the uniform distribution over the sphere $\mathbb{S}^{d-1}$.

We check the conditions of Theorem 2.1 in turn: It is straightforward to check that the risk function $R(\theta)$ is minimized at $\theta_0 = v$ and that $R(\theta)$ is $(\sqrt{d} + 1)$-Lipschitz, so that Conditions 1 and 2 hold. Checking Condition 3 is more involved, but it is a straightforward consequence of Gaussian concentration of measure, that Condition 3 holds with $t_0 = \sqrt{d/n}$, $c_1 = 2$ and $c_2 = 1/32$. See Lemma B.3 in the Supplementary Material.

Let us now determine sufficient values of $\beta$ and $n$ such that $\|\tilde{\theta}_n - \theta_0\| \leq t$ with probability at least $1 - \gamma$. To this end, let $W, W' \sim \mathcal{N}(0, 1)$ and $W$ be independent of $W'$, then we may write the minimum excess risk as

$$\alpha(t) = \mathbb{E} \log \sigma(W + 1) - \mathbb{E} \log \sigma(W' + 1 - t^2/2). \tag{3.1}$$

---

[1] As $\theta$ is effectively the normal vector for the separating hyperplane for the data, we need only to consider $\theta \in \mathbb{S}^{d-1}$ as rescaling $\theta$ will not affect the classifier after thresholding.

Note that this is a one-dimensional Gaussian integral and does not depend on $d$. To visualize this integral, we have included a graph of $\alpha(t)$ in the Supplementary Material, see Figure 1. Next, note that, as shown in Lemma B.1 in the Supplementary Material, for all $d > 2$ and all $0 < t < 2$,

$$-\log(\pi(B_t(\theta_0))) \lesssim d \log h(t \wedge 1),$$

where $h(t) = (1 - t^2/4)/(t\sqrt{1 - t^2/4} - t^2/4)$. Combining this fact with Theorem 2.1 gives the following result.

**Theorem 3.1.** *For the triple $(\ell, \mu, \pi)$ as in the described two component mixture model, all $d > 2$ and $t > 0$, $\|\tilde{\theta}_n - \theta_0\| \le t$ with probability at least $1 - \gamma$ provided that*

$$n \gtrsim \frac{\log(1/\gamma) \vee d}{\alpha(t)^2} \qquad \beta \gtrsim \frac{\log(1/\gamma) \vee d \log\left(h(\alpha(t)/8\sqrt{d})\right)}{\alpha(t)},$$

*where $\alpha$ is given in equation 3.1.*

One way to interpret Theorem 3.1 is as follows. For fixed $t$ and $\gamma = e^{-d}$, we have that $\log h(\alpha(t)/8\sqrt{d}) = O(\log d)$. Theorem 3.1 then says that taking $\beta \gtrsim d \log d/\alpha(t)$ guarantees an error level of $t$ with success probability $1 - e^{-d}$, provided that the number of samples satisfies

$$n \gtrsim d/\alpha(t)^2.$$

By way of comparison, the standard result for logistic regression is the PAC learning bound (Hanneke, 2016). One can show that in this case, the PAC learning bound can be written in terms of the minimum excess risk, which says that the number of samples needed to produce an estimator with error level $t$ with probability at least $1 - e^{-d}$, is given by

$$n \gtrsim d/\alpha(t)^2.$$

Comparing the bounds shows that if $\beta \gtrsim d \log d/\alpha(t)$, then drawing a sample from the Gibbs measure does not increase the sample complexity of the estimator. Lastly, the bound given on $\beta$ is tight up to logarithmic factors in $d$, see Remark 6.2.

## 4 Robust Regression

For our remaining examples, we focus on problems that are non-convex. One standard problem of this type is robust regression. Suppose that we observe $n$ i.i.d. data points, $\{(X_i, Y_i)\}_{i=1}^n$, such that $Y_i = X_i^\top \theta_0 + \epsilon_i$, where $X_i \sim \mathcal{N}(v, \sigma_x^2 I)$, $\epsilon_i \sim Cauchy(0, 1)$ and $\theta_0$ is some unknown, fixed vector of regression coefficients. In this case, the standard Ordinary Least Squares procedure will not perform well: if $\ell$ is the squared error loss, then for all $\theta \in \mathbb{R}^d$, $R(\theta) = \infty$. Similarly, using the Huber loss function, commonly used in robust statistics due to its convexity and robustness, will also result in a population risk that is infinite everywhere. A popular approach to resolving this issue is to work with the Tukey loss (Gross et al., 1973), also known as Tukey's biweight function

$$\ell(\theta, (X, Y)) = \begin{cases} \frac{\kappa^2}{6}\left(1 - \left[1 - \left(\frac{X^\top \theta - Y}{\kappa}\right)^2\right]^3\right) & \left\|X^\top \theta - Y\right\| < \kappa \\ \kappa^2/6 & \left\|X^\top \theta - Y\right\| \ge \kappa, \end{cases} \tag{4.1}$$

where $\kappa > 0$ is a hyperparameter. Under the Tukey loss, $R(\theta)$ is no longer degenerate. However, the Tukey loss is not convex. It may thus be difficult to compute the corresponding empirical risk minimizer. Let us then consider the performance of $\tilde{\theta}_n$ as an estimator of the minimizer of $R(\theta)$.

We begin by checking the conditions of Theorem 2.1. It is immediate that the risk function $R(\theta)$ is minimized at $\theta_0$ and is Lipschitz with constant $L = 2\kappa(\sqrt{d} \vee \|v\|)$ so that Conditions 1–2 are satisfied. To check Condition 3, we rely on the fact that the class of loss functions $\mathscr{F} = \{\ell(\theta, \cdot): \theta \in \mathbb{R}^d\}$ has the property $\mathrm{VC}(\mathscr{F}) = d + 1$, where VC denotes the Vapnik–Chervonenkis dimension of the class of functions $\mathscr{F}$, see Lemma B.2 in the Supplementary Material. Observe that in this case, the empirical risk is a bounded empirical process

with finite VC-dimension, and so we can apply Talagrand's inequality (Talagrand, 1994). This observation implies that Condition 3 is satisfied with $t_0 = 0$, $c_1 = (cn/d)^d$ and $c_2 = 72/\kappa^2$, where $c > 0$ is a universal constant, see equation B.1 and related discussion in the Supplementary Material for more detail. Now that the conditions of Theorem 2.1 have been checked, we now compute the minimum excess risk. Suppose we take $\pi = \mathcal{N}(\eta, \rho^2 I)$. First, observe that if $W \sim \mathcal{N}(0,1)$ and $\epsilon, \epsilon' \sim Cauchy(0,1)$ where $\epsilon$ is independent of $\epsilon'$, then

$$\alpha(t) = \frac{\kappa^2}{6}\left[\mathbb{E}\left(\left[1 - \frac{\epsilon^2}{\kappa^2}\right]^3 \mathbb{1}\left\{|\epsilon| < \kappa\right\} - \left[1 - \frac{(W\sigma_x^2 t + \epsilon')^2}{\kappa^2}\right]^3 \mathbb{1}\left\{|W\sigma_x^2 t + \epsilon'| < \kappa\right\}\right)\right].$$

Again, this is a relatively simple expression which does not depend on the dimension. Lastly, using elementary properties of the normal distribution, we have that if $\rho \geq 1/4$, then for all $t \leq \rho$, we have that

$$-\log \pi(B_t(\theta_0)) \lesssim \frac{\|\theta_0 - \eta\|^2}{\rho^2} + d\log\left(\frac{\rho}{t} \vee d\right),$$

see the proof of Lemma 15 of Ramsay et al. (2024) for more details. We are now in a position to apply Theorem 2.1, which gives the following result.

**Theorem 4.1.** *For the triple $(\ell, \mu, \pi)$ as in the described robust regression problem and all $t > 0$ it holds that $\|\tilde{\theta}_n - \theta_0\| \leq t$ with probability at least $1 - \gamma$ provided that*

$$n \gtrsim \kappa^2 \frac{\log(1/\gamma) \vee d(\log(\frac{\kappa}{c\alpha(t)}) \vee 1)}{\alpha(t)^2} \qquad and \qquad \beta \gtrsim \kappa \frac{\log(1/\gamma) \vee \left[\|\theta_0 - \eta\|^2/\rho^2 + d\log\left(\frac{\kappa\rho(d\vee\|v\|)}{\alpha(t)}\right)\right]}{\alpha(t)}.$$

Note that $\kappa$ is typically chosen to be $O(1)$ (Gross et al., 1973; Kafadar, 1983), and so suppose $\kappa = O(1)$. If in addition, $t = O(1)$, $\|v\|$ is polynomial in $d$ and $\|\theta_0 - \eta\|/\rho = O(\sqrt{d})$, then choosing the inverse temperature such that $\beta = O(d\log d)$ implies the sample complexity is $O(d\log d)$. Under the Cauchy error model, the sample complexity of the robust regression estimator is linear in $d$. Standard minimax theory provides a lower bound on the sample complexity of sub-Gaussian regression with identity covariance of $O(d)$. Therefore, we cannot hope to do much better $O(d)$ in the more challenging case of Cauchy errors. Note that the same analysis can be applied to nonlinear regression, where the conditional mean is given by $f(X, \theta)$, and the class $\mathscr{G} = \{f(\cdot, \theta); \theta \in \Theta\}$ satisfies $\mathrm{VC}(\mathscr{G}) < \infty$.

## 5 Spiked matrix and tensor models

Spiked matrix models (SMMs) and tensor models are popular statistical models used for a wide variety of inference and compression tasks (Johnstone, 2001; Richard & Montanari, 2014; Anandkumar et al., 2014). In these settings, however, the relevant loss functions are non-convex, which makes both information theoretic analysis and computation challenging. Despite this, there have been many breakthroughs in these areas over the past two decades using a variety of sophisticated techniques from random matrix theory and spin glass theory, e.g., (Johnstone, 2001; Baik et al., 2005; Benaych-Georges & Nadakuditi, 2011; El Alaoui et al., 2020; Jagannath et al., 2020; Perry et al., 2020). As a result, much is known about these models and the goal of this section is not to present new results concerning spiked matrix and tensor models, but to demonstrate that Theorem 2.1 can be used to quickly obtain information theoretic bounds for certain difficult problems rather easily.

We begin by analyzing several spiked matrix models via Theorem 2.1, under various popular assumptions on the unknown models, namely the spherical, Rademacher, and sparse settings. We can summarize these as follows, let $E \subset \mathbb{S}_\tau^{d-1}$ for some fixed $\tau > 0$, where $\mathbb{S}_\tau^{d-1} = \{x \in \mathbb{R}^d : \|x\| = \tau\}$. Suppose we have observed $n$ i.i.d. random matrices $\{A^{(i)}\}_{i=1}^n$ such that for some $\theta_0 \in E$ and all $i = 1, \ldots, n$,

$$A^{(i)} = W^{(i)} + \lambda\theta_0\theta_0^\top$$

where $\lambda > 0$, and each $W^{(i)}$ is a $d \times d$ matrix whose elements are independent and satisfy $W_{jk}^{(i)} = W_{kj}^{(i)} \sim \mathcal{N}(0, 1 + \mathbb{1}\{j = k\})$. In these problems, the goal is to produce an estimate of $\theta_0$. For $(\theta, A) \in E \times \mathbb{R}^{d\times d}$,

Table 1: Values of key quantities for different spiked matrix models. Note that if we let $Ber(p)$ be the Bernoulli measure with success probability $p$, then for $p, q \in (0,1)$ KL$(p, q)$ is the Kullback-Leibler divergence of $Ber(p)$ with respect to $Ber(q)$. Note that $f(t) = \alpha(t)/\lambda$ and $g$ is a bound on $-\log \pi(B_{\alpha(t)/8L}(\theta_0))$. See Theorem 5.1 and its proof in the Supplementary Material for more details on $f$ and $g$.

| | Classical | Rademacher | Sparse |
|---|---|---|---|
| $L$ | $2\lambda$ | $2\lambda d^{3/2}$ | $2\lambda\tau^3$ |
| $E$ | $\mathbb{S}^{d-1}$ | $\{\pm 1\}^d$ | $\{0,1\}^d \cap \mathbb{S}_\tau^{d-1}$ |
| $f(t)$ | $t^4(1-t^2/4)$ | $4\lceil t^2/2 \rceil (d - \lceil t^2/2 \rceil)$ | $\lceil t^2/2 \rceil (2\tau^2 - \lceil t^2/2 \rceil)$ |
| $g(d,\tau,t)$ | $d \log h(t^4(1-t^2/4)/16 \wedge 1)$ | KL$(t^8/d^4, 1/2) \cdot d \log d$ | $\tau^2 \log d$ |

the appropriate loss function is $\ell(\theta, A) = -\theta^\top A \theta$. Specific choices of $E$ recover well-known problems: taking $E = \mathbb{S}^{d-1}$ gives the classical spherical setting, taking $E = \{\pm 1\}^d$ gives the Rademacher prior and taking $E = \{0,1\}^d \cap \mathbb{S}_\tau^{d-1}$ gives the sparse models.

We now turn to checking the conditions of Theorem 2.1. First, it is easy to see that $R(\theta)$ is minimized at $\theta = \theta_0$ and so Condition 1 is satisfied. Further, $R(\theta) = -\lambda(\theta^\top \theta_0)^2$ which is $2\lambda\tau^3$-Lipschitz. For a matrix $V$ let $\|V\|$ be the operator norm of $V$ and let $\overline{W} = n^{-1} \sum_{i=1}^n W^{(i)}$. Condition 3 holds by Gaussian concentration of measure, i.e., noting that $\sqrt{n}\overline{W}$ and $W_1$ are identically distributed, we have that there exists universal constants $C, c > 0$ such that for $t \geq C\sqrt{d/n}$,

$$\Pr\left(\|\overline{W}\| \geq t\right) \leq e^{-cnt^2}.$$

Now, we are in a position to apply Theorem 2.1. Note that for $t \in [0, \tau]$, the minimum excess risk is given by

$$\alpha(t) = \lambda\tau^4 - \sup_{\{\theta \in E: \ \|\theta - \theta_0\| > t\}} \lambda(\theta^\top \theta_0)^2.$$

Observe that for each SMM, we can write $\alpha(t) = \lambda f(t)$. Values of $f$ for specific SMMs can be seen in Table 5. For each of these problems, we may take $\pi$ to be uniform on $E$. Lastly, by plugging in the specific values of $\pi$ for each problem, we can work out a bound on $-\log \pi(B_{\alpha(t)/8L}(\theta_0))$, which we call $g(d, \tau, t)$. The values of $g$ for each problem are given in Table 5, for more details on their derivation, see the proof of Theorem 5.1. For $j \in \{sph, rad, spa\}$, let $\pi_j$ be the uniform measure on $E$ as in the classical, the Rademacher, and the sparse spiked matrix models, respectively and let $\tilde{\theta}_{n,j}$ denote a sample from equation 2.1, corresponding to the triple $(\ell, \mu, \pi_j)$ implied by for the classical, the Rademacher, and the sparse spiked matrix models, respectively.

**Theorem 5.1.** *For all $j \in \{sph, rad, spa\}$, for each triple $(\ell, \mu, \pi_j)$, there are positive functions $f, g$ such that for all $t > 0$ and all $d > 2$ it holds that $\|\tilde{\theta}_{n,j} - \theta_0\| \leq t$ with probability at least $1 - \gamma$, provided that*

$$n \gtrsim \frac{\log(1/\gamma) \vee d}{\lambda^2 f(t)^2}, \qquad \beta \gtrsim \frac{\log(1/\gamma) \vee g(d, \tau, t)}{\lambda f_j(t)}. \tag{5.1}$$

Specific values of $g$ and $f$ are given in Table 5. Much of the SMM literature has focused on the case where $n = 1$, e.g., see Johnstone (2001); Perry et al. (2018); Gamarnik et al. (2021) and the references therein. One important question is that of weak recovery: How small can $\lambda$ be such that there is an estimator $\hat{\theta}$ of $\theta_0$ which has non-trivial correlation with $\theta_0$ as $d \to \infty$? A well-known result, known as the BBP transition, show that the smallest value of $\lambda$ for which we can weakly recover $\theta_0$ is $\sqrt{d}$. We see that reflected in Theorem 2.1. Indeed, taking $n = 1$ in equation 5.1 gives an upper bound of $O(\sqrt{d \log d})$ on the smallest $\lambda$ for which an estimate weakly recovers $\theta_0$. Another example can be taken from the sparse SMM. Suppose we assume that $\tau^2 = \rho d$, which implies that $d/\tau^2 = 1/\rho$. Gamarnik et al. (2021) show that the smallest value of $\lambda$ for which we can weakly recover $\theta_0$ is $O(\sqrt{-\log \rho}/\tau)$, which again matches the upper bound implied by equation 5.1 in Theorem 2.1.

We may also extend these results to Tensor principal component analysis. Specifically, define $W^{(1)}, \ldots, W^{(n)} \in (\mathbb{R}^d)^k$ such that each element $i_1, \ldots, i_k$ of each tensor $j$ is such that $W_{i_1,\ldots,i_k}^{(j)} \sim \mathcal{N}(0,1)$,

and all elements are independent. Suppose instead that for some $\lambda > 0$ and $\theta_0 \in \mathbb{S}^{d-1}$, we observe $A^{(i)} = W^{(i)} + \lambda \theta_0^{\otimes k}$ where $x^{\otimes k}$ denotes the $k$th tensor of a vector. In this problem, we have that

$$R(\theta) = -\lambda \langle \theta^{\otimes k}, \theta_0^{\otimes k} \rangle \geq -\lambda,$$

with equality at $\theta = \theta_0$. Therefore, $R(\theta)$ is minimized at $\theta = \theta_0$ and so Condition 1 is satisfied. Further, the risk function is $k\,\lambda$-Lipschitz and so Condition 2 is also satisfied. For a tensor, $V$ let $\|V\|$ be the operator norm of $V$. Condition 3 holds again by Gaussian concentration. It follows from (Ben Arous et al., 2019, see Lemma 4.7) that for all $t > 0$

$$\Pr\left(\|\overline{W}\| \geq t\right) \leq e^{d \log k - nt^2/8}. \tag{5.2}$$

The next step is to compute the minimum excess risk, which, in this case, for $t < \sqrt{2}$,

$$\alpha(t) = \lambda \left(1 - (1 - t^2/2)^k\right).$$

It is again intuitive to set the prior to be the uniform measure on $\mathbb{S}^{d-1}$. We are now in a position to apply Theorem 2.1, which results in the following

**Theorem 5.2.** *For the triple $(\ell, \mu, \pi)$ as in the described Tensor PCA model and all $k \in \mathbb{N}$, $t > 0$, $d > 2$ and $\lambda > 0$, we have that $\|\tilde{\theta}_n - \theta_0\| \leq t$ with probability at least $1 - \gamma$ provided that*

$$n \gtrsim \frac{\log(1/\gamma) \vee d \log k}{\lambda^2 (1 - (1 - t^2/2)^k)^2},$$

$$\beta \gtrsim \frac{\log(1/\gamma) \vee d \log h((1 - (1 - t^2/2)^k)/8k)}{\lambda(1 - (1 - t^2/2)^k)}.$$

Since our bounds are general, they are necessarily not sharp. Nevertheless, they can be seen to match the correct order of growth as shown by other authors: Letting $\lambda^*$ be the threshold below which weak recovery is impossible, it is easy to see that Theorem 2.1 implies that $\lambda^* \leq K\sqrt{d \log k}$, for a constant $K > \sqrt{2}$ which matches scaling in $k$ and $d$ obtained by Perry et al. (2020).

# 6 Technical details

We now turn to proving the technical results in the manuscript.

*Proof of Theorem 2.1.* The proof is in the same spirit as that of Ramsay et al. (2024). For brevity, let $D_{n,t} = \left\{ \max_{\theta \in A^*} d(\tilde{\theta}_n, \theta) > t \right\}$, and for any $y > 0$, let $E_{n,y} = \left\{ \sup_{\theta \in \Theta} |\widehat{R}_n(\theta) - R(\theta)| < y \right\}$. Note that from Condition 3, we have that for any $y > t_0$ it holds that

$$\Pr\left( \max_{\theta \in A^*} d(\tilde{\theta}_n, \theta) > t \right) \leq c_1 e^{-c_2 n y^2} + \Pr\left( E_{n,y} \cap D_{n,t} \right). \tag{6.1}$$

By definition, we have that

$$\frac{1}{\beta} \log \Pr\left( E_{n,y} \cap D_{n,t} \right) = \frac{1}{\beta} \log \int_{E_{n,y}} \frac{\int_{B_t^c(A^*)} \exp\left(\beta\, \widehat{R}_n(\theta)\right) d\pi}{\int_{\mathcal{X}} \exp\left(\beta\, \widehat{R}_n(\theta)\right) d\pi} d\mu. \tag{6.2}$$

On $E_{n,y}$ it holds that

$$\exp\left(-\beta\, R(\theta)\right) \exp(-\beta\, y) \leq \exp\left(-\beta\, \widehat{R}_n(\theta)\right) \leq \exp\left(-\beta\, R(\theta)\right) \exp\left(\beta\, y\right). \tag{6.3}$$

We can then apply equation 6.3 to the right-hand side of equation 6.2, which yields

$$\frac{1}{\beta} \log \Pr\left( D_{n,t} \cap E_{n,y} \right) \leq \frac{1}{\beta} \log \frac{\int_{B_t^c(A^*)} \exp\left(-\beta R(\theta)\right) d\pi}{\int_{\mathcal{X}} \exp\left(-\beta R(\theta)\right) d\pi} + 2y. \tag{6.4}$$

The next step is to bound the denominator below. For any $r > 0$, Condition 2 implies that

$$\frac{1}{\beta}\log\int_{\mathcal{X}}\exp\left(-\beta R(\theta)\right)d\pi \geq \frac{1}{\beta}\log\int_{B_r(A^*)}\exp\left(-\beta R(\theta)\right)d\pi$$

$$= -\inf_{\theta\in\Theta} R(\theta) + \sup_{\theta_0\in A^*}\frac{1}{\beta}\log\int_{B_r(A^*)}\exp\left(-\beta(R(\theta)-R(\theta_0))\right)d\pi$$

$$\geq -\inf_{\theta\in\Theta} R(\theta) - L\cdot r + \beta^{-1}\log\pi(B_r(A^*)).$$

Maximizing the right-hand side of the above, and recalling the definition of equation 2.3, yields that

$$\frac{1}{\beta}\log\int_{\mathcal{X}}\exp\left(-\beta R(\theta)\right)d\pi \geq -\inf_{\theta\in\Theta} R(\theta) - \psi(\beta)/\beta.$$

We can then plug this lower bound into equation 6.4, resulting in

$$\frac{1}{\beta}\log\Pr\left(D_{n,t}\cap E_{n,y}\right) \leq \frac{1}{\beta}\log\int_{B_t^c(A^*)}\exp\left(-\beta R(\theta)\right)d\pi + \inf_{\theta\in\Theta} R(\theta) + \psi(\beta)/\beta + 2y$$

$$\leq -\inf_{\theta\in B_t^c(A^*)} R(\theta) + \inf_{\theta_0\in\Theta} R(\theta_0) + \frac{1}{\beta}\log\pi(B_t^c(A^*)) + \psi(\beta)/\beta + 2y$$

$$= -\alpha(t) - I(t)/\beta + \psi(\beta)/\beta + 2y,$$

where the last line follows from the definitions of $\alpha(t)$ and $I(t)$. Rewriting the last inequality gives that

$$\Pr\left(D_{n,t}\cap E_{n,y}\right) \leq \exp\left(-\beta\alpha(t) - I(t) + \psi(\beta) + 2\beta y\right). \tag{6.5}$$

Now, setting $y = f(t,\beta) = \alpha(t)/2 \vee I(t)/2\beta$ and plugging equation 6.5 into equation 6.1 results in

$$\Pr\left(\max_{\theta\in A^*} d(\tilde{\theta}_n,\theta) > t\right) \leq c_1 e^{-nc_2 f(t,\beta)^2} + \Pr\left(D_{n,t}\cap A_{n,f(t,\beta)}\right)$$

$$\leq c_1 e^{-nc_2 f(t,\beta)^2} + e^{-\beta\alpha(t) - I(t) + \psi(\beta) + 2\beta f(t,\beta)}. \tag{6.6}$$

Now, note that

$$\alpha(t) + I(t)/\beta - f(t,\beta) \geq \alpha(t)/2 + I(t)/2\beta.$$

Plugging this inequality into equation 6.6 results in

$$\Pr\left(\max_{\theta\in A^*} d(\tilde{\theta}_n,\theta) > t\right) \leq c_1 e^{-nc_2 f(t,\beta)^2} + e^{-\beta\alpha(t)/2 - I(t)/2 + \psi(\beta)}. \qquad \square$$

The following lemma is useful for proving the remaining theorems.

**Lemma 6.1.** *If the Conditions of Theorem 2.1 hold then, for all $t > 0$ satisfying $\alpha(t) \vee I(t)/\beta \geq 2t_0$ we have that $\sup_{A^*} d(\tilde{\theta}_n,\theta_0) \leq t$ with probability at least $1 - \gamma$ provided*

$$n \geq \frac{\log(e/2c_1\gamma)}{c_2\alpha(t)^2}, \qquad and \qquad \beta \gtrsim \frac{\log(1/\gamma) \vee -\log\pi(B_{\alpha(t)/8L}(A^*))}{\alpha(t)}. \tag{6.7}$$

*Proof.* The aim is to show that $\Pr\left(\max_{\theta_0\in A^*} d(\tilde{\theta}_n,\theta_0) \geq t\right) \leq \gamma$. Focusing on the left-hand term, when

$$n \geq \frac{\log(e/2c_1\gamma)}{c_2\alpha(t)^2}.$$

we have that

$$c_1 e^{-c_2 n\left[\frac{1}{\beta}I(t)\vee\alpha(t)\right]^2/4} \leq \gamma/2.$$

Next, note that together, $\psi_\pi(\beta) \le \beta\alpha(t)/4$ and $\beta \ge 4\log(1/\gamma)/\alpha(t)$ imply that

$$e^{-\beta\alpha(t)/2 - I(t)/2 + \psi_\pi(\beta)} \le \gamma/2.$$

Thus, it remains to find a condition on $\beta$ such that $\psi_\pi(\beta) \le \beta\alpha(t)/4$. To this end, recall that by definition $\psi_\pi(\beta) = \min_{r>0}[\beta \cdot L \cdot r - \log \pi(B_r(A^*))]$. Taking $r = \alpha(t)/8L$ in the right-hand expression above gives $\psi_\pi(\beta) \le \beta\alpha(t)/8 - \log \pi(B_{\alpha(t)/8L}(A^*))$. It is then enough to have

$$\beta \ge 8\frac{-\log \pi(B_{\alpha(t)/8L}(A^*))}{\alpha(t)}. \qquad \square$$

*Remark* 6.2 (Sharpness of bound). We note here that our bound on the inverse temperature required is in a sense sharp. To see this, suppose for simplicity that we are interested in a high-dimensional inference task where we seek to infer an unknown vector on the unit sphere $\theta_0 \in \mathbb{S}^{d-1}$. Suppose that our loss satisfies $\ell(\theta, X) \le 1$ and that $\pi$ is the uniform measure on $\mathbb{S}^{d-1}$. This set-up captures,e.g., the PCA and logistic regression settings above, and our arguments show that the sampling approach works provided $\beta \lesssim d$. This is sharp: By an elementary bound, $\nu_\beta(A) \le e^{2\beta}\pi(A)$ for any $A \subseteq \mathbb{S}^{d-1}$. Taking $A = \{\theta : \langle\theta, \theta_0\rangle > t\}$, a standard concentration bound yields $\pi(A) \lesssim e^{-c(t)d}$ for some $c(t) > 0$ so that $\nu_\beta(A) \lesssim e^{2\beta - c(t)d}$ Thus if $\beta = o(d)$, $\nu_\beta(A) \to 0$, which would contradict the performance of the sampling-based estimator

## Acknowledgments

The authors acknowledge the support of the Natural Sciences and Engineering Research Council of Canada (NSERC). Cette recherche a été financée par le Conseil de recherches en sciences naturelles et en génie du Canada (CRSNG), [DGECR-2023-00311, DGECR-2020-00199].

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
