

Figure 1: Minimum excess risk $\alpha(t)$ for the example in Section 3

## A    Additional graph of $\alpha(t)$ from Section 3

Figure 1 displays a graph of $\alpha(t)$ for the example in Section 3.

## B    Auxiliary lemmas

**Lemma B.1.** *Define*

$$h(r) = \frac{1 - r^2/4}{r\sqrt{1 - r^2/4} - r^2/4}.$$

*Suppose that $\pi$ is the uniform measure on $\mathbb{S}^{d-1}$. For all $d > 2$ and all $r > 0$,*

$$-\log \pi(B_r(\theta_0)) \lesssim d \log\left(h(r \wedge 1)\right).$$

*Proof.* In order to compute $\pi(B_r(\theta_0))$, we must first compute the surface area of the set $A = \{x\colon \|x\| = 1, \|x - v/s\| \le r\}$. Let Beta$(a, b)$ be the Beta function and let $I_x(a, b)$ be the regularized incomplete Beta function. If $r < \sqrt{2}$, then the set $A$ is a spherical cap with radius $R = r\sqrt{1 - r^2/4}$ and height $H = r^2/2$, which has surface area:

$$\text{SA}(A) = \frac{\pi^{d/2} R^{d-1}}{\Gamma(d/2)} \cdot I_{H(2R-H)/R^2}\left(\frac{d-1}{2}, 1/2\right).$$

If $r > \sqrt{2}$, letting $H' = 2 - r^2/2$, it follows that $A$ has surface area:

$$\text{SA}(A) = \frac{\pi^{d/2} R^{d-1}}{\Gamma(d/2)}\left(1 - I_{H'(2R-H')/R^2}\left(\frac{d-1}{2}, 1/2\right)\right).$$

We focus on the case where $r < \sqrt{2}$, it then follows that

$$\pi(B_r(\theta_0)) = \frac{1}{2} \cdot I_{H(2R-H)/R^2}\left(\frac{d-1}{2}, \frac{1}{2}\right).$$

Now, define

$$g(r) = H(2R - h)/R^2 = \frac{r\sqrt{1 - r^2/4} - r^2/4}{1 - r^2/4}.$$

Note that $g$ is decreasing toward 0 as $r \to 0$. One can show that for $d > 2$, we have that

$$I_{g(r)}\left(\frac{d-1}{2}, \frac{1}{2}\right) \ge \frac{2\sqrt{\pi}}{d-1}\left(\frac{d}{2} - 1\right)^{-1/2} g(r)^{d/2 - 1/2}.$$

Therefore, for

$$\pi(B_r(\theta_0)) \geq \frac{1}{2} \cdot \frac{2\sqrt{\pi}}{d-1} \left(\frac{d}{2} - 1\right)^{-1/2} g(r)^{d/2-1/2}.$$

Taking the negated log gives that

$$-\log \pi(B_r(\theta_0)) \lesssim d\log(1/g(r)).$$

The result follows from the fact that $h(r) = 1/g(r)$. □

**Lemma B.2.** *Let $g_\kappa$ be Tukey's biweight function. Then, the VC dimension of the class of functions given by $\{g_\kappa(X^\top\theta - Y) \colon \theta \in \mathbb{R}^d\}$ is $d+1$.*

*Proof.* First, note that the class of real-valued linear functions on $\mathbb{R}^d$, denoted by $\mathscr{F}$, satisfies $\mathrm{VC}(\mathscr{F}) = d+1$. We can then write $(X^\top\theta - Y)^2 = ((X^\top\theta - Y) \vee (Y - X^\top\theta))^2$. Now, the subgraphs of $\{(X^\top\theta - Y) \vee (Y - X^\top\theta) \colon \theta \in \mathbb{R}^d\}$ are the union of the subgraphs of $\{Y - X^\top\theta \colon \theta \in \mathbb{R}^d\}$ and $\{X^\top\theta - Y \colon \theta \in \mathbb{R}^d\}$, which is simply the set of subgraphs of linear functions. Thus, $\mathrm{VC}(\{(X^\top\theta - Y) \vee (Y - X^\top\theta) \colon \theta \in \mathbb{R}^d\}) = d+1$. Next, note that $g_\kappa$ can be written as

$$g_\kappa(\theta) = \frac{\kappa^2}{6}\left(1 - \left[1 - \frac{(X^\top\theta - Y)^2}{\kappa^2}\right]^3\right) \wedge \frac{\kappa^2}{6},$$

which is a monotonic function on $\mathbb{R}^+$. It follows from the permanence properties of VC-dimension, e.g., see Lemma 7.12 of (Sen, 2018), that

$$\mathrm{VC}(\{g_\kappa(X^\top\theta - Y) \colon \theta \in \mathbb{R}^d\}) = \mathrm{VC}(\mathscr{F}) = d+1.$$ □

**Lemma B.3.** *Let $X, Y$ be as in Section 3. Then, for all $t \geq \sqrt{d/n}$ we have that*

$$\Pr\left(\sup_{\theta \in \mathbb{S}^{d-1}} \left|\frac{1}{n}\sum_{i=1}^n \log\sigma(X_i\langle X_i, \theta\rangle) - \mathbb{E}_\mu \log\sigma(Y\langle X, \theta\rangle)\right| \geq t\right) \leq 2e^{-nt^2/32}.$$

*Proof.* Let $Z = (Z_1, \ldots, Z_n)$, where $Z$ is defined in Section 3 and let $Z' \sim \mathcal{N}(v, I)$ be independent of $Z$. Define

$$g_\theta(Z) = \frac{1}{n}\sum_{i=1}^n \log\sigma(\langle Z_i, \theta\rangle) - \mathbb{E}\log\sigma(\langle Z', \theta\rangle).$$

Note that $g_\theta$ are all $1/\sqrt{n}$-Lipschitz functions of $Z$. It follows that $\sup_{\theta \in \mathbb{S}^{d-1}} g_\theta$ is also a $1/\sqrt{n}$-Lipschitz function of $Z$. A log–Sobolev inequality on finite-dimensional Gaussian space gives that

$$\Pr\left(\sup_{\theta \in \mathbb{S}^{d-1}} g_\theta(Z) \geq \mathbb{E}\sup_{\theta \in \mathbb{S}^{d-1}} g_\theta(Z) + t\right) \leq e^{-nt^2}.$$

We can further write

$$\mathbb{E}\sup_{\theta \in \mathbb{S}^{d-1}} g_\theta(Z) = \mathbb{E}\sup_{\theta \in \mathbb{S}^{d-1}} g_\theta(Z) - \sup_{\theta \in \mathbb{S}^{d-1}} g_\theta(0) + \sup_{\theta \in \mathbb{S}^{d-1}} g_\theta(0) \leq \frac{1}{\sqrt{n}}\mathbb{E}\|Z\| = \sqrt{\frac{d}{n}}.$$

As a result, for $t \geq \sqrt{d/n}$ we have that

$$\Pr\left(\sup_{\theta \in \mathbb{S}^{d-1}} \left|\frac{1}{n}\sum_{i=1}^n \log\sigma(Y_i\langle X_i, \theta\rangle) - \mathbb{E}_\mu \log\sigma(Y\langle X, \theta\rangle)\right| \geq t\right) \leq 2e^{-nt^2/32}.$$

□

*Proof of Theorem 3.1.* The conditions required for Theorem 2.1 have been checked in Section 3, and so we can apply Lemma 6.1. In this case, we have that $t_0 = 4\sqrt{d/n}$ and so it suffices to have $n \geq 8d/\alpha(t)$. Furthermore, in this case, $c_1, c_2$ are universal constants, and so the equation 6.7 becomes $n \gtrsim (\log(1/\gamma) \vee d)/\alpha(t)^2$. The next step is to bound

$$- \log \pi(B_{\alpha(t)/8(\sqrt{d} \vee s)}(\theta_0)).$$

An application of Lemma B.1 with $r = \alpha(t)/8(\sqrt{d} \vee s)$ gives the result. $\square$

*Proof of Theorem 4.1.* Let $\mathscr{F} = \{6\ell(\theta, \cdot, \cdot)/\kappa^2 \colon \theta \in \mathbb{R}^d\}$. It follows from Lemma B.2 that $\mathrm{VC}(\mathscr{F}) = d + 1$ and that for any $f \in \mathscr{F}$ we have that $\|f\| \leq 1$. Thus, an application of Talagrand's inequality (Talagrand, 1994, Theorem 1.1), see also (Sen, 2018, Theorem 7.11), or (Kosorok, 2008, Theorem 9.3) implies that there is some universal $c > 0$ such that for all $n \geq 1$ and for all $t > 0$, it holds that

$$\Pr\left(\sup_{\theta \in \mathbb{R}^d} |\hat{R}_n(\theta) - R(\theta)| > t\right) \leq e^{d \log(cn/d) - 72nt^2/\kappa^4}. \tag{B.1}$$

Thus, Condition 3 is satisfied with $t_0 = 0$, $c_1 = (cn/d)^d$ and $c_2 = 72/\kappa^4$. The remaining conditions of Theorem 2.1 were checked in Section 4, thus, we can apply Lemma 6.1. The condition on $n$ reduces to the following

$$n \gtrsim \kappa^2 \frac{\log(1/\gamma) \vee d(\log(\frac{\kappa}{c\alpha(t)}) \vee 1)}{\alpha(t)^2}.$$

Then, Lemma B.4 of (Ramsay et al., 2024) (stated below for convenience) gives that

$$- \log \pi(B_{\alpha(t)/8\kappa(\sqrt{d} \vee \|v\|)}(\theta_0)) \lesssim \frac{\|\theta_0 - \eta\|^2}{\rho^2} + d \log\left(\frac{\kappa\rho(d \vee \|v\|)}{\alpha(t)}\right).$$

Combining this inequality with equation 6.7 yields the condition

$$\beta \gtrsim \kappa \frac{\log(1/\gamma) \vee \left[\|\theta_0 - \eta\|^2/\rho^2 + d \log\left(\frac{\kappa\rho(d \vee \|v\|)}{\alpha(t)}\right)\right]}{\alpha(t)}. \qquad \square$$

*Proof of Theorem 5.1.* The conditions of Theorem 2.1 were checked in Section 5, thus, we can apply Lemma 6.1. Note that $t_0 = C\sqrt{d/n}$, which implies that the bounds in Lemma 6.1 hold for $n \gtrsim d/\alpha(t)^2$. Next, the inequality equation 6.7 reduces to $n \gtrsim \log(1/\gamma)/\alpha(t)^2$. It remains to bound $-\log\pi(B_t(\theta_0))$ for each prior. For $\pi$ uniform on the unit sphere, applying Lemma B.1 yields that equation 6.7 reduces to

$$\beta \gtrsim \frac{\log(1/\gamma) \vee d\left(\log h(t^4(1 - t^2/4)/16)\right)}{\lambda t^4(1 - t^2/4)}.$$

In Rademacher prior PCA, we take $\pi$ to be uniform on $\{\pm 1\}^d$. For such a prior, we have that each vector can be observed with probability $2^d$. Computing $B_r(\theta_0)$ requires counting the number of vectors within $r$ of $\theta_0$. The quantity $\lfloor r^2/2 \rfloor$ gives the maximum Hamming distance between $x \in B_r(\theta_0)$ and $\theta_0$. Let $Z \sim Binomial(d, 1/2)$ and $\mathrm{KL}(p, q)$ be the Kullback–Leibler divergence of $Binomial(1, p)$ with respect to $Binomial(1, q)$. Following Ash (1990), we have that there exists a universal constant $c > 0$ such that

$$\pi(B_{\alpha(t)/8L}(\theta_0)) = \pi(B_{t^4/16d^{3/2}}(\theta_0)) \geq \Pr\left(Z \leq ct^8/d^3\right) \gtrsim \frac{1}{\sqrt{d}} e^{-d\,\mathrm{KL}(t^8/d^4, 1/2)}.$$

This results in

$$- \log \pi(B_{t^4/16d^{3/2}}(\theta_0)) \lesssim d\,\mathrm{KL}(t^8/d^4, 1/2) \log d.$$

In this case, we have that

$$\beta \geq C \frac{\mathrm{KL}(t^8/d^4, 1/2) \cdot d \log d}{4\lambda\lceil t^2/2\rceil(d - \lceil t^2/2\rceil)}.$$

Combining this with the bound $\beta \geq Cd/\alpha(t)$ yields

$$\beta \geq C \frac{(\mathrm{KL}(t^8/d^4, 1/2) \cdot d \log d) \vee d}{4\lambda \lceil t^2/2 \rceil (d - \lceil t^2/2 \rceil)}.$$

In the sparse PCA case, we have that $\tau^2$ is the number of 1s in the vector. Each vector in $E$ may occur with probability $1/\binom{d}{\tau^2}$. Furthermore, it is easy to show that $\alpha(t)/8L \gtrsim \frac{t^2(\tau^2 \wedge (d-\tau^2))}{\tau^3}$. This implies that $\lfloor \frac{r^2}{2} \rfloor \gtrsim a_{t,\tau,d}$, with $a_{t,\tau,d} = \frac{t^4(\tau^2 \wedge (d-\tau^2))^2}{\tau^6}$. We have that

$$\pi(B_{\alpha(t)/8L}(\theta_0)) = \sum_{i=0}^{\lfloor \frac{r^2}{2} \rfloor} \frac{\binom{\tau^2}{i}\binom{d-\tau^2}{i}}{\binom{d}{\tau^2}} \geq \frac{\binom{\tau^2}{a_{t,\tau,d}}\binom{d-\tau^2}{a_{t,\tau,d}}}{\binom{d}{\tau^2}} \geq \left( \frac{\tau^2 \wedge (d-\tau^2)}{de} \right)^{2a_{t,\tau,d}+\tau^2 \wedge (d-\tau^2)}.$$

This results in

$$-\log \pi(B_{\alpha(t)/8L}(\theta_0)) \lesssim (a_{t,\tau,d} \vee \tau^2) \log d \lesssim \tau^2 \log d.$$

This bound yields the following condition on $\beta$ for sparse PCA:

$$\beta \gtrsim \frac{\log(1/\gamma) \vee \tau^2 \log d}{\lambda \lceil t^2/2 \rceil (2\tau^2 - \lceil t^2/2 \rceil^2)}. \qquad \square$$

*Proof of Theorem 5.2.* The conditions of Theorem 2.1 have been checked in Section 5 and applying Lemma 6.1 yields the following bound on $n$

$$n \gtrsim \frac{\log(1/\gamma) \vee d \log k}{\lambda^2 (1 - (1 - t^2/2)^k)^2}.$$

It now remains to bound $-\log \pi(B_{\alpha(t)/8k\lambda}(\theta_0))$, which follows from applying Lemma B.1. As a result, equation 6.7 reduces to

$$\beta \gtrsim \frac{\log(1/\gamma) \vee d \log h((1 - (1 - t^2/2)^k)/8k)}{\lambda(1 - (1 - t^2/2)^k)}. \qquad \square$$

We restate the following result from (Ramsay et al., 2024) for convenience.

**Lemma B.4** (Ramsay et al. (2024))**.** *If $\pi = \mathcal{N}(\eta, \rho^2 I)$ with $\rho \geq 1/4$, then for all $E \subset \mathbb{R}^d$, all $d > 2$ and all $R \leq \rho$ it holds that*

$$-\log \pi(B_R(E)) \lesssim \frac{d(E,\eta)^2}{\rho^2} + d \log \left( \frac{\rho}{R} \vee d \right). \tag{B.2}$$