# OpenReview forum: "An elementary concentration bound for Gibbs measures arising in statistical learning theory"
_TMLR — Accepted by TMLR_

### Review · Reviewer_cku6 · 2024-09-27

**Summary Of Contributions:**

**Summary Of Contributions**

1. This paper derives a concentration bound for the Gibbs measure whose log-likelihood is the product of the empirical risk and the inverse temperature. The concentration upper bound depends on the calibration function $\psi_{\pi}$, the minimal excess risk $\alpha$ and the rate function $I$, which further depend on the number of samples $n$, the inverse temperature $\beta$ and the prior $\pi$. By quantifying the dependency of $\psi_{\pi},\alpha,I$ on $n,\beta,\pi$, the concentration bound can be applied to determine how large the inverse temperature and the number of samples should be to guarantee that a sample from the Gibbs measure is within a distance t of a minimizer of the population risk.

2. The paper applies the aforementioned approach to analyze three canonical risk minimization examples: classification for a two component mixture model, robust regression and spiked matrix/tensor models.

**Audience:**

Yes

**Broader Impact Concerns:**

No ethical concerns for this paper.

**Claims And Evidence:**

Yes

**Requested Changes:**

1. As the author mentioned in the paper, the concentration bound is general and may not imply a sharp bound in specific problems. In the three examples, it might be good to also introduce what the state of the art bounds on $n$ and $\beta$ look like. That would help to understand how good this concentration method actual is when applied to different problems.

2. In Condition 3, it might be good to also emphasize $c_1,c_2$ can depend on $d,n$.

3. The prior $\pi$ is chosen to be the uniform distribution in example 1. This should be stated explicitly in the main paper.

4. In examples 2 and 3, it might be better to also introduce how the rate function $I$ is estimated, just like what's done in example 1.

**Strengths And Weaknesses:**

**Strength**

1. The concentration result in the paper is established under relative mild conditions, which makes the method easy to be applied to varies risk minimization tasks.

**Weaknesses**

1. Although the authors mentioned the improvement of the concentration result from the one in [1], how this improvement is achieved methodologically is not explained. In my opinion, it will make the paper more insightful if a discussion on the methodology of proving the concentration bound can be added.

---

> ### Author Response · Authors · 2024-12-03
>
> **Comparison to Ramsay, et al., 2024:** The goal of this paper is to illustrate how one can apply the main technical lemma in Ramsay, et al., 2024 to a broader class of problems in statistical learning beyond differential privacy. Our main result has weaker conditions than that paper: Conditions 1 and 2 are the same in both papers. Condition 3 in Ramsay, et al., 2024 implies Condition 3 in our paper by Talagrand’s inequality for empirical processes. By relaxing this condition, the range of problems to which this lemma applies is dramatically increased. For example, Condition 3 in Ramsay, et al., 2024 is not satisfied for the spiked matrix and tensor application, nor the classification application, but Condition 3 in our paper is satisfied.
>
> We have clarified this in Section 1.
>
> **Proof:** We have not included further discussion of the methodology of the proof, as the proof is only one page long.
>
> **Sharp bounds:** We have done this in each example.  For Example 1, see the paragraph immediately after Theorem 3.1, for Example 2, see the paragraph immediately after Theorem 4.1, for Example 3, see the paragraphs immediately after Theorem 5.1 and Theorem 5.2. See also, Remark 6.2.
>
> **Changes 2,3:** We have made these changes.
>
> **Change 4:** In our applications, we bound $-\log\pi(B_r(A^*))$ using the properties of the prior selected, which is straightforward, since these are specific measures with densities or mass functions we can write down. We have added a note in each section on how this is bounded, with a pointer to the correct place for more details.

---

### Review · Reviewer_CfUN · 2024-10-11

**Summary Of Contributions:**

This paper utilizes the recently developed concentration bound by Ramsay et al. (2024) to address the following question: How large should the inverse temperature $\beta$, and the sample size n, be to ensure that a sample from the Gibbs measure is within a distance t of a minimizer of the population risk? Additionally, three canonical examples are provided, focusing on the classification of two-component mixture models, robust regression, and spiked matrix and tensor models.

**Audience:**

Yes

**Claims And Evidence:**

Yes

**Requested Changes:**

1. Recent advancements in the information-theoretic characterizations of the Gibbs algorithm should be noted:

Aminian, Gholamali, Yuheng Bu, Laura Toni, Miguel Rodrigues, and Gregory Wornell. "An exact characterization of the generalization error for the Gibbs algorithm." Advances in Neural Information Processing Systems 34 (2021): 8106-8118.

Bu, Yuheng. "Towards Optimal Inverse Temperature in the Gibbs Algorithm." In 2024 IEEE International Symposium on Information Theory (ISIT), pp. 2257-2262. IEEE, 2024.

Zou, Xinying, Samir M. Perlaza, Iñaki Esnaola, and Eitan Altman. "Generalization analysis of machine learning algorithms via the worst-case data-generating probability measure." In Proceedings of the AAAI Conference on Artificial Intelligence, vol. 38, no. 15, pp. 17271-17279. 2024.

2. Can the proposed method be applied to understand ERM by letting $\beta$ approach infinity? Including such a discussion after theorem 2.1 would certainly enhance the quality of the paper.

Minor comments:

1.	The metric d() in the parameter space should be introduced when defining the Lipschitz condition. More generally, can the proof technique be extended to other metrics, provided they satisfy the Lipschitz condition?

2.	What is the prior distribution used in Section 3? Also, there is no Lemma A.1 in the paper, referring to (Lemma 6.1)?

**Strengths And Weaknesses:**

Strengths:

Studying sample complexity bounds and bounds on the inverse temperature to ensure that small population risk is a fundamental problem in learning theory. Additionally, since the Gibbs algorithm can be viewed as the asymptotic limit of SGLD, it has practical applications for guiding hyperparameter selection.

Weaknesses:

A more detailed characterization of the minimum excess risk function $\alpha(t)$ is needed. Currently, it is not immediately clear how the gap in the population risk depends on the number of samples and the inverse temperature. While I understand that $\alpha(t)$ is problem-dependent, could you either work out Equation (3.1) or provide some figures to help the reader better understand this dependence?

---

> ### Author Response · Authors · 2024-12-03
> **Review response CfUN**
>
> **Work out equation (3.1) or provide a graph on \alpha(t):**  It is problem dependent, and we did not simplify (3.1) further, but we have now included a graph in the appendix.
>
> **Additional literature:** We have now cited this literature in the introduction.
>
> **Discuss $\beta\to \infty$:** Yes, our bound applies in that setting. We have added this discussion in Remark 2.3 following Theorem 2.1.
> Specifically, when $\beta$ becomes large, we expect the error of $\tilde\theta_n$ to be similar to that of the true minimizer of the empirical risk, say $\hat\theta_n$. This is reflected in (2.3), where, under a weak condition, the term in (2.3) pertaining to using a draw from the Gibbs measure, rather than the minimizer of the empirical risk, disappears. The condition is that $\beta\to\infty$ faster than $\psi(\beta)$, which holds when the prior has sufficient density on the minimizing set and the risk is smooth enough. Sufficient density and smooth enough can be made explicit in the specific examples. Going further, taking $n\to\infty$, we have that (2.3) implies that $\tilde\theta_n$ is weakly consistent for $\theta$.
>
> **Minor changes:** We have made these changes. We have added the metric $d$ before equation (2.1). The prior in Section 3 is now stated. Note that Lemma A.1 is part of the Appendix. We have made this clearer in the manuscript.

---

### Review · Reviewer_qCrq · 2024-11-24

**Summary Of Contributions:**

This paper presents a concentration bound for the Gibbs measure, which can be applied in the setting of statistical learning under certain conditions. From a technical perspective, this result is an extension of a previous result in a different setting. The authors also demonstrated this result in three learning problems.

**Audience:**

Yes

**Claims And Evidence:**

Yes

**Requested Changes:**

See Weakness for details.

The Major Weaknesses are critical for the acceptance of this work.
If these are addressed/cleared, I can bear with the rest minor weaknesses.

**Strengths And Weaknesses:**

Strength:
(1) At an initial reading, all of the statements and proofs appear to be sound.

(2) The authors demonstrate the applicability of the proposed bounds on three learning problems.

Weakness:
Major:
(1) The authors claim that this work relaxed the conditions in Ramsey, et al., 2024, which is part of the main contribution, however, it is not clear whether or not this is true. There is no introduction/discussion on Ramsey, et al., 2024 and no comparison of the conditions between these two. One should not be required to read another paper to understand this work. Moreover, Condition 3 requires sub-Gaussian property on the unknown distribution $\mu$ and I am not entirely convinced this is a very relaxed condition.

(2)This paper seeks to explore the relationship between the performance of a Gibbs sampler and the choice of sample size, prior, and inverse temperature. If the goal is to provide deeper understanding, further analysis of the right-hand side of equation (2.3) would be valuable. For instance, how should the assumption $\alpha(t) \vee I(t) \ge 2t_0$ be interpreted beyond its mathematical convenience for the proof? Additionally, what practical insights does equation (2.3) offer for What insights does (2.3) bring for choosing $\beta$? There are many studies on selecting the inverse temperature for the Gibb measure, such as “Towards Optimal Inverse Temperature in the Gibbs Algorithm” by Yuheng Bu. A comprehensive study of the properties of the Gibbs algorithms can be also found in ‘‘Empirical Risk Minimization with Relative Entropy Regularization’’ by Samir Perlaza, et al. More discussion on the insights should be provided.

(3)The main result of this paper—Theorem 2.1—presents a PAC-Bayes bound specifically for the Gibbs measure, an area of active research. Notably, “User-friendly Introduction to PAC-Bayes Bounds” by Pierre Alquier and other related references offer important context. However, this paper does not mention prior work on PAC-Bayes bounds, which weakens its positioning within the literature. Also, analysis/discussion on the tightness of this bound would be appreciated.

(4) The paper is missing a Conclusion section at the end.

Minor:
(1)The mathematical notations in this paper lack clarity. For example, in Condition (3) and Equation (2.3), the sub-Gaussian inequality does not explicitly specify which random variables it pertains to, making the statements ambiguous and harder to interpret. Providing precise definitions and clear references to the random variables involved would greatly enhance the readability and rigor of the paper.

---

> ### Author Response · Authors · 2024-12-03
> **Review response qCrq**
>
> **Comparison to Ramsay, et al., 2024:** The goal of this paper is to illustrate how one can apply the main technical lemma in Ramsay, et al., 2024 to a broader class of problems in statistical learning beyond differential privacy. In particular, our main result has weaker conditions than that paper: Conditions 1 and 2 are the same in both papers. Condition 3 in Ramsay, et al., 2024 implies Condition 3 in our paper by Talagrand’s inequality for empirical processes. By relaxing this condition, the range of problems to which this lemma applies is dramatically increased. For example, Condition 3 in Ramsay, et al., 2024 is not satisfied for the spiked matrix and tensor application, nor the classification application, but Condition 3 in our paper is satisfied.
> **We have clarified this in Section 1.**
>
> It is important to note that **Condition 3 does not require sub-Gaussianity** of the data distribution. For example, in Section 4, the results are applied to a setting where the data distribution has no moments, specifically robust regression with Cauchy errors.
> Furthermore, we can weaken Condition 3 further, to say,
> “There exists $t_0\coloneqq t_0(n,\Theta)>0$ and a sequence of real-valued functions $\omega_n$, which may depend on $\Theta$, such that for all $t\geq t_0$, and $n\geq 1$, it holds that $\Pr(\sup_{\theta\in \Theta}|\widehat R_n(\theta)-R(\theta)|\geq t)\leq \omega_n(t),$”
> for which we get the bound
>
> $$\Pr(\max_{\theta\in A^*}d(\tilde{\theta}_n,\theta)\geq t)\leq  \omega_n((I(t)/2\beta)\vee (\alpha(t)/2))  +\exp(-\beta\alpha(t)/2-I(t)/2+\psi(\beta)) ,$$
>
>
> if one wishes to avoid this assumption. For all our applications, the rate is sub-Gaussian, so we left the bound as is.
>
> **Lower bound on $\alpha(t) \vee I(t)/\beta$:** The role/interpretation of this bound depends on the problem. For instance, in all our applications, it ends up being not important –
>
> 1.	Sometimes $t_0=0$ and this is always satisfied.
> 2.	Other times, $t_0=f(n,d)$ and implies that $n> g(d,\beta ,t)$. However, when we compute sample complexity bounds, we get that $n>h(d,\beta,t)> g(d,\beta ,t)$, and so the bound $\alpha \vee I(t)/\beta>t_0$ is satisfied anyways if the lower bound on the sample complexity is satisfied.
>
> We have added a few sentences summarizing this after Theorem 2.1 and preceding Remark 2.2.
>
>
> **What insights does (2.3) bring for choosing $\beta$?:** Theorem 2.1 can be used to prove bounds on $\beta$ under which the estimate $\tilde\theta_n$ achieves optimal sample complexity.  In (2.3) the upper bound has are two terms, the sampling error term and the energy-entropy term. To see this, observe that the sampling error term yields an upper bound on the sample complexity of the estimator. This upper bound on the sample complexity is often the optimal sample complexity up to logarithmic terms, see the example applications in the subsequent sections. Then, to achieve optimal sample complexity, we need to choose $\beta$ such that the energy-entropy term goes to zero faster than the sampling error term. It suffices to bound $\beta$ as a function of $n, t$ and $\Theta$ such that both $\psi(\beta)\lesssim \beta \alpha(t)$ and $\beta \gtrsim n\alpha(t)$.
> We had a paragraph summarizing this after Theorem 2.1, but we have expanded it now and made this point clearer.  This is the second paragraph following Theorem 2.1.
>
> **Additional literature on Information theory of Gibbs measures:**  We have now cited this literature in the introduction.
>
> **Compare to PAC-Bayes:** We respectfully disagree with the reviewer on this point. Theorem 2.1 is not a PAC Bayes bound. To clarify, PAC-Bayesian bounds are traditionally high probability bounds on the generalization error of the sampled parameter. Our result bounds the distance between the sampled parameter and the true minimizer of the population risk, with high probability.
> These bounds are not equivalent functionally nor conceptually. They can be related under additional assumptions on the population risk itself, for example, strong convexity of the population risk or the specific case where the population risk is equal to the distance between the sampled parameter and the true minimizer of the population risk.
>
> We have now added a pointer to the PAC-Bayes literature in Section 1.
>
> **Conclusion:** We respectfully disagree with the reviewer on this point. The goal of this paper is to state a concentration inequality and its proof, and to provide examples of how it can be applied. It is not common for papers of this type to have conclusions.

---

> > ### Comment · Reviewer_qCrq · 2024-12-05
> > **Reply to the authors**
> >
> > Thank you for providing a detailed response to my questions. I just want to understand better. Regarding the new sentence in the paper: "Condition 3 says that the empirical risk concentrates uniformly around the population risk at
> > a sub-Gaussian rate." From the $\sup_{\theta}$, to my understanding, for all $\theta$, $\hat{R}_n(\theta) - R(\theta)$ is required to be a subgaussian random variable. Isn't it a requirement for the randomness carried out by the data samples/data distribution?
> >
> > Thanks for the comments on PAC-Bayes, I agree with the authors that this is not by definition a traditional PAC-Bayes result.

---

> > > ### Author Response · Authors · 2024-12-05
> > >
> > > Happy to explain. In applications, the interplay between the data distribution and Condition 3 depends deeply on the chosen loss function.
> > >
> > > For example, suppose we have an i.i.d. sample $X_1,\ldots,X_n$ whose mean is $E(X_1)=\mu<\infty$, and we want to do mean estimation in one dimension through empirical risk minimization.
> > >
> > > If we choose the Tukey loss function, viz. $$ \ell(\theta,X)=
> > >     \begin{cases}
> > >     \frac{\kappa^2}{6}\left(1-\left[1-\left(\frac{X-\theta}{\kappa}\right)^2\right]^3\right) & \quad|X-\theta|< \kappa \\\\
> > >     \kappa^2/6 & \quad|X-\theta|\geq \kappa
> > >     \end{cases}  , $$
> > > then, observe that for all $\theta,x\in\mathbb{R}$, $\ell(\theta,x)<\kappa^2/6$, i.e., the loss is bounded for any pair $(\theta,x)$. This implies immediately that $\widehat R_n(\theta)<\kappa^2/6$, which implies that for all $\theta$, $| \widehat R_n(\theta)-R(\theta)|<\kappa^2/3$. Therefore, $| \widehat R_n(\theta)-R(\theta)|$ is a sub-Gaussian random variable, however, $X_1,\ldots,X_n$ can be drawn from any distribution with finite mean.
> > >
> > > On the other hand, taking the absolute error loss, and letting $\mathbf{X}=(X_1,\ldots,X_n)$, we would require that $\sup_{\theta}|R_n(\theta)-R(\theta)|=\sup_{\theta}| ||\mathbf{X}-\theta\mathbb{1}_n||_1/n- E(|X-\theta|)| $ be sub-Gaussian for all $\theta$, which would required a sub-Gaussian tail bound on the data distribution.

---

### Author Response · Authors · 2024-12-03
**Overall response and description of changes**

Dear all, thank you for your kind and careful reviews.

We have updated the manuscript, with important changes highlighted in red text for your convenience.

In general, **we have mainly edited the introduction and the text following Theorem 2.1**.

**The introduction changes include:** a comparison to relevant literature that was missed, a comparison to PAC Bayes bounds, and an explicit explanation of how our bound is a generalization of the bound given by Ramsay, et al., 2024 .

**The changes following Theorem 2.1 together make a clearer and deeper explanation of Theorem 2.1. These include:** how it can be used to derive upper bounds on the sample complexity of a draw from the Gibbs measure, what happens when the inverse temperature goes to infinity and a discussion of the requirement $\alpha(t) \vee I(t)/\beta>t_0$.

**We have also responded to each of your concerns individually below.**

---

### Decision · Action_Editor_o414 · 2025-02-17

**Recommendation:** Accept as is

**Comment:**

This paper presents a concentration bound for Gibbs-type measures, extending previous work by Ramsay et al. The work generalizes the previous results, using weaker conditions, and offers a broadening of scope of applicability.

The technical claims and proofs appear to be correct, as verified by the reviews based on the revision and discussion with the authors. The original version lacked some technical details, but these have been provided in the updated manuscript and the reviewers are happy with the updates. In private discussion, one reviewer suggested that the comparison with Ramsay et al. (2024) in the introduction could be presented in a way that is more accessible to a general audience - I encourage you to do so in the final version.

**Audience:**

Yes, this would be of interest to some of the TMLR audience

**Claims And Evidence:**

This paper presents a technical bound for Gibbs-type measures, extending work by Ramsay et al (2024) in a different setting. While the initial manuscript lacked certain key details, these have been provided in the updated manuscript and verified by the reviewers.